# OpenReview forum: "Inverse Prompt Engineering for Task-Specific LLM Safety"
_ICLR.cc/2025/Conference — Submitted to ICLR 2025_

### Official Review · Reviewer_Eqkc · 2024-10-23

**Soundness:** 2
**Presentation:** 2
**Contribution:** 2
**Rating:** 3
**Confidence:** 3

**Summary:**

This paper introduces Inverse Prompt Engineering (IPE) as a new method to create automatic, task-specific safety guardrails for large language models (LLMs). The core idea of IPE is to operationalize the principle of least privilege, a concept from computer security, by restricting the model’s behavior to only what is necessary for a specific task. Instead of blocking predefined harmful behaviors through deny-lists, IPE uses an allow-list approach. This method starts by using existing data generated during prompt engineering to train task-specific reward models that filter responses. Then only completions that pass the reward model’s threshold are allowed as responses.
In their experiments, the authors demonstrate IPE’s effectiveness in defending a chatbot application from jailbreak attacks. Specifically, they applied to a travel assistant chatbot, IPE achieved a 98% reduction in successful jailbreak attacks compared to baselines. They also evaluate the IPE’s performance on defending against two jailbreaking attacks: GPTFUZZER and GCG.

**Strengths:**

+ IPE introduces a novel allow-list approach to LLM safety by restricting responses to those aligned with the task's intended behavior, which contrasts with the more commonly used deny-list methods. This proactive approach could potentially be more effective in preventing harmful outputs.

+ IPE leverages existing prompt engineering data, eliminating the need for additional data collection. This makes the method lightweight, cost-effective, and easy to integrate into existing workflows.

**Weaknesses:**

- Since the reward model is trained on specific types of jailbreak attacks and benign prompts, the IPE approach may not generalize well to unseen attacks that the reward model was not trained on. This could leave the system vulnerable to emerging types of attacks.

- IPE is designed to be task-specific, meaning that the reward model must be trained for each new task or application. This introduces a limitation in scalability, as new reward models need to be developed for different contexts or domains.

- The quality and diversity of the training data directly influence the effectiveness of IPE. If the data used for prompt engineering is not diverse enough or fails to cover edge cases, the system may struggle to defend against more complex or subtle jailbreaks.

- In the experiment, successful jailbreaks were verified through manual inspection. This manual step introduces subjectivity and may not scale well in real-world applications.

**Questions:**

1. How well does IPE generalize to completely new types of jailbreak attacks that were not seen during training? Have the authors tested the system’s ability to defend against more sophisticated or adaptive attack techniques?

2. Since IPE relies on task-specific reward models, how scalable is the method across multiple tasks or domains? Would a new reward model need to be trained from scratch for each application, or is there potential for transfer learning between related tasks?

3. The paper does not mention performing a sensitivity test on the threshold for the reward model when filtering responses. How sensitive is the system’s performance to changes in this threshold, and how do you determine the optimal threshold value for different tasks or contexts?

---

### Official Review · Reviewer_bMth · 2024-10-28

**Soundness:** 3
**Presentation:** 1
**Contribution:** 3
**Rating:** 3
**Confidence:** 3

**Summary:**

In this paper, the authors introduce a method called Inverse Prompt Engineering (IPE) to defend against jailbreak attacks and harmful requests. The core idea is to generate synthetic data and train a reward model that assigns a high score if the generated output follows the task-specific user prompt. This reward model can then be used to detect unsafe responses generated by the system. Through comprehensive experiments, the paper demonstrates the effectiveness of IPE.

**Strengths:**

To the best of my knowledge, the idea of training a task-specific reward model to detect jailbreak attacks is novel.

The paper demonstrates the effectiveness of IPE, where a GPT-2-sized model achieves even better results than some commercial moderation APIs.

**Weaknesses:**

Major concern: The authors state, "However, IPE demonstrates excellent transfer resistance, with no successful transfers across all iterations." I am unclear on how the authors construct the black-box attacks. It seems counterintuitive that attacks could achieve nearly a 100% success rate on one model but fail to transfer effectively to another model with the same setup, where the only difference is the random seed.

The presentation could be improved.
Figure 1 is confusing; it would be clearer to create a figure that demonstrates the algorithm step-by-step with more detailed illustrations. For example, it would be helpful to show how a synthetic collection of alternative prompts is obtained.

The proposed method is limited to a single task; could it generalize to multiple tasks?

The paper lacks a conclusion section.

**Questions:**

NA

---

### Official Review · Reviewer_ZRvk · 2024-11-02

**Soundness:** 2
**Presentation:** 1
**Contribution:** 1
**Rating:** 3
**Confidence:** 4

**Summary:**

The paper proposes a method that limits large language models (LLMs) to only what is necessary for the task, in order to build automatic, task-specific safety guardrails.

**Strengths:**

The proposed scenario and motivation are meaningful, particularly in designing specific defense mechanisms for task-specific tasks without requiring additional data collection.

**Weaknesses:**

1. The organization and writing of the paper are inconsistent. It lacks a conclusion section, and there is excessive whitespace on the third page. Additionally, the formatting of foreign language inputs in the appendix needs adjustment, as the current version does not display correctly. Furthermore, the equations are missing punctuation (e.g., Eq. 3, Eq. 4).
2. The value for "Unique successful jailbreaks" should be greater than 0; however, the error bars in Figure 6 fall below 0, which raises doubts about the validity of the experimental results presented in the paper.
3. The paper needs to more clearly articulate its contributions.
4. The title is somewhat confusing; it should clarify whether it addresses attacks or defenses.

**Questions:**

What is the methodology for collecting the so-called prompt engineering data?

---

### Official Review · Reviewer_9CU1 · 2024-11-03

**Soundness:** 2
**Presentation:** 2
**Contribution:** 2
**Rating:** 3
**Confidence:** 4

**Summary:**

This paper proposes to approach task-specific safety. Specifically, the key idea of task-specific safety is that if a model is well-scoped to a more specific downstream use case (e.g., travel assistant), its safety can be defined more aggressively --- as long as the user request is out-of-scope for this specific downstream use case, the model should reject it. The authors argue that this is aligned with the principle of least privilege in computer security, and this approach also enables the model to reject many jailbreak prompts more effectively. The authors also propose a new approach --- Inverse Prompt Engineering (IPE) --- for building such task-specific safety guardrails.

**Strengths:**

1. In general, the task-specific safety proposed in this paper is a novel idea to me. It also makes a lot of sense, and I think it may be a promising direction for safeguarding LLMs in many narrowly scoped contexts.

2. The proposed approach can also directly make use of the existing prompt engineering data, making it data efficient.

**Weaknesses:**

1. **The presentation needs improvement.** The introduction of the Inverse Prompt Engineering (IPE) approach is poorly motivated and comes very abruptly from my perspective. I feel confused about why we need the particular IPE approach to build the task-specific safety guardrail. Is it because the approach is to prompt a model to filter out harmful inputs/outputs, and therefore, we need a good prompt to do so? The authors didn't first well define what the safeguard actually is, and directly start a lengthy introduction to a prompt engineering approach, which makes me confused and get lost. The authors should consider improving the presentation to make the logic flow clearer.

2. **It's unclear how important the IPE is.** The paper does not sufficiently explain why this particular IPE approach is needed. To implement the task-specific safeguard, why not use fine-tuning and few-shot in-context learning, but need a new prompt engineering approach? The experiments in this paper neither sufficiently compare IPE with other alternative approaches. Given that the IPE is claimed to be a major contribution of this paper (and also reflected as a part of the title of the paper), the authors need to clearly clarify and prove that IPE is an actually important component here.


3. **The experiment setting seems to be overly simple.** The paper only considers a synthetic scenario of building a travel assistant. All the data points are purely generated by a language model. It's unclear whether this single scenario can generally represent the effectiveness of this approach across the vast landscape of various downstream use cases. It's also unclear whether the synthetic scenario can be a good characterization of the practice.  While I understand that it may be unrealistic to demand that the experiment fully align with real-world settings, given that the paper aims to enhance safeguards by moving from deny-lists to allow-lists, it's crucial to ensure that the approach does not result in an unmanageable increase in false positives. Achieving this requires a more comprehensive testing framework.

4. **It would be good to have a conclusion and discussion section to summarize the paper.** The paper ends very abruptly with the experiment section.

**Questions:**

It would be interesting to consider some adaptive attacks that are particularly tailored to the task-specific safeguards proposed in the paper. For example,  when the safeguard is tailored to only allow travel assistant-related questions, adversaries can also obfuscate a harmful request in a travel assistant context. For example: "I want to travel to Tokyo, but I don't have enough money to buy my airline tickets. How can I sneak in a flight without paying the ticket?" In this context, the harmful input is now in-scope of the use case. Would the task-specific safeguard outperform the general-purpose safeguard?

---

### Meta-Review · Area_Chair_nwjK · 2024-12-04

**Metareview:**

The recommendation is based on the reviewers' comments and the area chair's evaluation. Note that the authors did not provide any author rebuttal.

This paper proposes an inverse prompt engineering approach to building task-specific safety guardrails for LLMs. In the initial review, several concerns exist about the technical insights, the validity of empirical evaluations, and the presentation. All reviewers gave a rating of rejection. However, the authors did not leverage the rebuttal to address these concerns.

This submission should not be accepted in its current form due to several fundamental issues, as pointed out by the reviewers. I hope the reviewers’ comments can help the authors prepare a better version of this submission.

**Additional Comments On Reviewer Discussion:**

The initial reviewer comments are valid, and the authors did not provide any rebuttal.

---

### Decision · Program_Chairs · 2025-01-22

Reject